# PA200-Mediated Proteasomal Protein Degradation and Regulation of Cellular Senescence

**DOI:** 10.3390/ijms25115637

**Published:** 2024-05-22

**Authors:** Pei Wen, Yan Sun, Tian-Xia Jiang, Xiao-Bo Qiu

**Affiliations:** 1State Key Laboratory of Natural Medicines, China Pharmaceutical University, Nanjing 211198, China; 3121030151@stu.cpu.edu.cn (P.W.); 1020222647@cpu.edu.cn (Y.S.); 2Ministry of Education Key Laboratory of Cell Proliferation & Regulation Biology, College of Life Sciences, Beijing Normal University, 19 Xinjiekouwai Avenue, Beijing 100875, China

**Keywords:** proteasome, protein degradation, PA200, Blm10, PSME4, aging, senescence

## Abstract

Cellular senescence is closely related to DNA damage, proteasome inactivity, histone loss, epigenetic alterations, and tumorigenesis. The mammalian proteasome activator PA200 (also referred to as PSME4) or its yeast ortholog Blm10 promotes the acetylation-dependent degradation of the core histones during transcription, DNA repair, and spermatogenesis. According to recent studies, PA200 plays an important role in senescence, probably because of its role in promoting the degradation of the core histones. Loss of PA200 or Blm10 is a major cause of the decrease in proteasome activity during senescence. In this paper, recent research progress on the association of PA200 with cellular senescence is summarized, and the potential of PA200 to serve as a therapeutic target in age-related diseases is discussed.

## 1. Introduction

Cellular senescence is accompanied by a progressive decline in the function of cells with a durable cell-cycle arrest of previously replication-competent cells [1]. Cellular senescence can be triggered by multiple genetic alterations induced by oxidative stress, DNA damage, or telomere shortening [2]. Senescent cells are characterized by five distinct features: irreversible cell cycle arrest, senescence-associated secretory phenotype (SASP), increased apoptosis resistance, enlarged cell morphology, and overexpression of lysosomal enzymes and the senescence-associated β-galactosidase [3]. Cell cycle arrest can be caused by inhibiting cell cycle progression through p16^INK4^ and/or activating cell cycle arrest through p53/p21 [4,5,6]. The SASP typically includes mRNA, chemokines, cytokines, extracellular vesicles (EVs), growth factors, proteases, and interleukins [7,8]. SASP affects nearby cells through paracrine signaling, inhibiting their transformation into tumor cells and inducing cellular senescence [9]. However, it can also promote tumor development [10]. Studies on the metabolism of senescent cells have shown that these cells exhibit changes in metabolism compared to proliferating cells [6,11]. This may cause senescent cells to exhibit an enlarged cellular phenotype [12]. Old mammalian cells are typically 2–3 times larger than young cells [13]. Cellular senescence is associated with not only genomic instability but also epigenetic changes, such as alterations in DNA methylation and histone modifications. Recently, emerging evidence suggests that epigenetic dysregulation is both a hallmark and a driver of aging. Restoring epigenetic integrity can reverse the aging phenotype [14,15]. The levels of the acetylated histone H4 at K16 (H4K16ac) are upregulated during senescence [16]. Histone 3 lysine 4 trimethylation (H3K4me3) is a marker of chromatin opening located on transcription start sites (TSSs) and is associated with not only active transcription but also senescence regulation [17]. Partial histone loss and the ensuing dysregulated transcription could be associated with aging [8,18,19,20].

Cellular senescence is often associated with the abnormal accumulation of proteins, probably caused by dysregulation of proteasomes, which catalyze the degradation of most cellular proteins [8,21]. While the 26S proteasome, which contains the 19S regulatory particle, promotes degradation of the ubiquitinated proteins, the PA200-proteasome promotes degradation of the acetylated core histones during cellular aging, transcription, somatic DNA repair, and spermiogenesis [22,23,24]. The activity of the 26S proteasome has been shown to decline during aging in various organisms [25,26]. The cellular levels of the 26S proteasome correlate with a longer replicative lifespan in yeast. Deletion of the 19S particle subunit Rpn4, a transcription factor for most subunits of the 26S proteasome [27], reduces yeast replicative lifespan [21]. Notably, loss of PA200 or its yeast ortholog Blm10 is the leading cause of the decline in proteasome activity during aging [28]. Thus, this review will focus on the recent progress in studies of the roles of PA200 in cellular senescence.

## 2. Overview of Proteasomes

Proteasomes are composed of a 20S catalytic core particle and one or two activators, including the 19S regulatory particle, the PA28 heptamers (including the PA28α/β heteroheptamer and the PA28γ homoheptamer), and PA200 (or its yeast ortholog Blm10) [29]. The proteasomal activators are located at one or both ends of the 20S particle.

### 2.1. The 20S Catalytic Particle

The 20S core particle (20S CP) is a cylindrical complex composed of two α-rings and two β-rings with seven subunits in every single ring [30]. One α-ring is present on the exterior of the cylinder, which controls the substrate entry and release of the degradation products. Two β-rings form a catalytic chamber with trypsin-like, caspase-like, and chymotryptic-like activities, where substrates are degraded (Figure 1A). These activities are conferred by the β1/PSMB6, β2/PSMB7, and β5/PSMB5 subunits. In hematopoietic cells and cells stimulated by IFN-γ, the catalytic subunits of the 20S particle are replaced by β1i (LMP2), β2i (MECL-1) and β5i (LMP7) subunits to form the immunoproteasome 20S (i20S) [31]. In testes, the testis-specific proteasome subunit α4s/PSMA8, which is essential for male fertility, promotes the proper formation of spermatoproteasomes, which harbor both PA200 and the 20S particle with constitutive catalytic subunits. In the adult testis of the α4s-deficient mice, PA200 binds not only the 20S particle with regular catalytic subunits but also the i20S particle [22,23,24].

The size of the 20S catalytic particle is about 700 kDa. The N-terminus of the α-subunits constructs two narrow axial gating channels, controlling the substrate entry and release of the degradation products. The α rings act as barriers for the degradation lumen, preventing the entry of non-degradable proteins into the lumen by mistake. α subunits can independently polymerize to form a ring, and their assembly is necessary for the formation of the β ring. The catalytic chamber, with a volume of about 84 nm^3^, is composed of 2 β-rings. The two sides are combined to form a “receiving chamber (antechambers)” with a volume of about 59 nm^3^. The combination of the two sides can be applied to control the substrate entry and release of the degraded products. A “receiving chamber” with a volume of about 59 nm^3^ is formed on each side of the catalytic chamber. The receiving chamber receives a large amount of substrate to be degraded [32].

### 2.2. The 19S Regulatory Particle

The 19S regulatory particle (19S RP) is a complex with more than 20 subunits. There are six ATPase (Rpt) subunits, which form a hexameric ring in the order Rpt1–Rpt2–Rpt6–Rpt3–Rpt4–Rpt5 in the 19S RP. The C-termini of all Rpt subunits, except Rpt6, bind to the α rings of the 20S particle [33]. Rpt subunits utilize the hydrolysis of ATP to drive the unfolding and entry of substrates into the 20S particle. Six non-ATPase (Rpn) subunits (Rpn9, Rpn5, Rpn6, Rpn7, Rpn3, and Rpn12) are assembled into a horseshoe-like structure to position Rpn11 above the AAA-ATPase module. Rpn1 and Rpn2 are linked by an extended connection to facilitate the coordination of the rotational movement of other Rpns. In addition, Rpn1 [34], Rpn10 [35], and Rpn13 [36] are the intrinsic ubiquitin receptors to catch ubiquitinated substrates (Figure 1B). The polyubiquitin chain has to be removed from the substrate by the 19S RP before substrate degradation [37]. There are three deubiquitylating enzymes associated with the 19S RP in mammals, including Rpn11, UCH37, and Usp14 [38,39,40], but only Rpn11 is constitutive [40,41,42,43]. Thus, the 19S regulatory particle recognizes ubiquitin-tagged substrates, unfolds substrates, removes ubiquitin chains, and regulates the entry of substrates into the 20S particle.

### 2.3. Proteasome Activators PA28α, PA28β and PA28γ

There are two isoforms of PA28-proteasomes (Figure 1C). One is the immunoproteasome, which consists of the heteroheptamer of proteasomal activators PA28α/PA28β and three inducible catalytic subunits β1i, β2i, and β5i to facilitate antigen presentation [44,45,46]. The other is the PA28γ-proteasome, which consists of the homoheptamer of the proteasomal activator PA28γ to degrade certain substrates [47,48,49]. The mechanism by which PA28γ regulates the activity of the 20S CP has undergone several updates. Initially, it was believed that PA28γ only targeted the trypsin-like (T-L) 20S β-catalytic site, acting as an activator of peptide substrate hydrolysis [49]. However, the result of a regulatory effect of PA28γ on 20S CP seems to be related to the purification method. Ammonium sulfate precipitation was used to purify recombinant PA28γ, and door-opening activity was observed [50], while the same phenomenon was observed for the other two purification methods [51,52]. While recombinant PA28γ was classically purified using ion exchange and size exclusion chromatography, N-terminally His-tagged recombinant PA28γ was purified using the Nickle resin. The transiently overexpressed N-terminally Flag-tagged PA28γ from a cell line showed that PA28γ specifically upregulated the resolution of the T-L peptide and did not exhibit 20S gate opening activity [49,53]. Interestingly, a single-site mutation of PA28γ at lysine-188 (K188E) indicates that 20S CP regulation changes from T-L—activation to gate opening [54]. Recently, Thomas and Smith developed a method to study PA28γ-activated proteasome activity using the α3ΔN 20S mutant. α3ΔN is a mutant proteasome species with a 10-residue N-terminal truncation of the α3 subunit. The α3 N-terminus stabilizes the closed state of the proteasome by interacting with α2 and α4 in the 20S pore. Thus, deletion of the α3 N-terminus results in a proteasome with an intrinsically open channel [55]. Their work demonstrated the function of PA28γ as an activator of the T-L protein hydrolysis site. This study also shows the first cryo-electron microscopy (cryo-EM) density map of the PA28γ-20S CP complex, which shows a tetrameric structure similar to other 11S-20S CP complexes [48]. In the same year, Chen et al. published the 3.4 Å structure of PA28γ, confirming the previous results [56]. The substrates of the PA28γ-proteasome are involved in the regulation of cell cycle, apoptosis, DNA damage, lipid metabolism, and other processes [57,58,59,60,61]. Studies on the physiological function of PA28γ have also shown its relevance in a variety of diseases, such as cancer [62,63,64], neurodegenerative diseases [65], hepatitis B infection [66], and COVID-19 [67].

### 2.4. The Proteasome Activator PA200/Blm10

The proteasome activator PA200 is a monomeric protein of 200 kDa (Figure 1D). The homologs of PA200 are also found in worms (*Caenorhabditis elegans*), plants (*Arabidopsis thaliana*), and yeast (*Saccharomyces cerevisiae*) [68]. Although there is a nuclear localization sequence, PA200 is detectable in both cytoplasmic and nuclear extracts [69,70]. PA200 primarily consists of 32 HEAT-like repeats, which form a dome-like structure attached to the 20S particle [70,71]. The C-terminal residues YYA (Tyr-Tyr-Ala) of PA200 cause an α-ring rearrangement, leading to a partial opening of the 20S gate [71,72]. The binding between PA200 and the 20S particle selectively activates its trypsin-like activity and slightly inhibits its caspase- and chymotryptic-like activities [72,73]. Phosphatidylinositol binds and regulates the catalytic activity of histone deacetylases [74,75]. There are two unique pore channels in PA200 formed by a large number of positively charged residues that bind phosphatidylinositol [71,73]. Whether the binding of phosphatidylinositol to PA200 regulates the recognition of the acetylated substrates might deserve to be tested.

The key functions of histones are to compact DNA and provide epigenetic regulation of transcription. Chromatic histones were generally thought undegradable in somatic cells until 2013, when PA200 was shown to promote proteasomal degradation of the core histones in an acetylation (rather than ubiquitination)-dependent manner during sperm spermiogenesis and somatic DNA repair [22]. The levels of histones in chromatin drop 20–40% in response to DNA damage [76]. The PA200-proteasome was also shown to degrade the acetylated core histones during DNA damage-induced replication stress [77]. The bromodomain (BRD) binds the acetylated lysine and is usually composed of a left-handed helix bundle formed by four alpha helices and two hydrophobic loops (Figure 1E). Both PA200 and Blm10 contain a BRD-like domain (BRDL), which is required for binding and degradation of the core histones [22,78]. However, the detailed mechanism underlying the interaction of PA200 with the core histones remains to be further explored.

## 3. PA200 Plays Important Roles in Preventing Cellular Senescence

Loss of PA200 or Blm10 is the leading cause of the decline in proteasome activity during aging [28]. The transcription factor Crt1 suppresses the expression of Blm10, but the proteasome subunit Rpn4, which transactivates most subunits of the 26S proteasome, promotes the transcription of Blm10 upon DNA damage. Contrary to the deletion of Rpn4, the deletion of Crt1 reduces core histone levels during aging and prolongs replicative lifespan [28].

### 3.1. PA200-Mediated Degradation of the Core Histones during Senescence

Partial depletion of histones in the genome has been shown to be closely related to senescence in yeast and human cells [79]. To study whether the PA200-mediated degradation of the core histones happens in non-replicating cells, a modified pulse-chase assay was developed to measure histone degradation in cultured cells by metabolically labeling proteins with a substitute of Met, azidohomoalanine (Aha). The PA200-proteasome promotes the acetylation-dependent degradation of the core histones in the non-replicating G1-arrested cells [24]. A genome-wide analysis of histone degradation (GAHD) was performed following sequencing DNA fragments purified together with histones after 2-h pulse labeling with Aha and chase in regular medium for 0 or 4 h. PA200 promotes degradation of the core histones primarily in regions of active transcription [24]. Chromatin immunoprecipitation (ChIP)-sequencing analyses suggest that PA200 is essential for the maintenance of histone marks in gene regions. Particularly, PA200 regulates the deposition of the transcriptionally active histone marks, including H3K4me3 and H3K56ac, a process inversely correlated with DNA methylation, which usually marks transcriptionally inactive regions in certain critical gene regions [80]. Recent studies have shown that H3K4me3 is associated with cellular senescence. Specifically, Spt-Ada-Gcn5 acetyltransferase (SAGA) complex-associated factor 29 (SGF29)-mediated phase separation creates a subcellular environment for H3K4me3 recognition in the promoter region, activating cell cycle protein-dependent kinase inhibitor 1A (CDKN1A, p21) and accelerating human stem cell senescence [81]. On the other hand, the recruitment of RNA polymerase II onto chromatins is positively correlated with this deposition of histone marks. Further, RNA sequencing results showed that deletion of PA200 promotes the senescence-promoting genes Bmp4, Cdkn1a (p21^Cip1^), and Cdkn2b (p15^INK4b^) and suppresses the expression of the senescence-suppressing genes Hmgb1 and Sod1. Meanwhile, deletion of PA200 was found to promote cellular senescence in primary mouse fibroblasts. Finally, the PA200-deficient mice display a range of aging-related phenotypes, including immune malfunction, anxiety-like behaviors, and a much shorter lifespan [24]. These results suggest that the aging-related phenotypes in PA200-deficient mice may be caused by the accumulation of “aged” histones with aberrant post-translational modifications. In another related study, the levels of asymmetric demethylation of histone H4 at arginine 3 (H4R3me2as) were shown to be negatively correlated with the interaction between PA200 and histone H4, while H4 degradation promotes senescence-associated gene transcription. Notably, anti-aging drugs (metformin, rapamycin, and resveratrol) can restore the levels of H4 much earlier than other senescence markers in response to the oxidant H_2_O_2_ treatment [82]. Thus, PA200 might extend the cell lifespan by degrading the core histones with aberrant histone marks and regulating the expression of senescence-related genes during senescence.

Similarly, Blm10 deletion markedly reduced degradation of the Flag-tagged H3 as analyzed in the non-replicating G1-arrested yeast. The degradation of the endogenous histones also shows a Blm10 dependence using a pulse-chase assay with metabolic Aha-labeling [83]. Although the levels of Blm10 in cells were lowered dramatically during senescence, senescence induces the transcriptional upregulation of Blm10 [28]. Overexpression or deletion of Blm10 regulates gene expression in senescent yeast much more than in young yeast. These genes are associated with transcription, amino acid metabolism, nucleotide metabolism, carbohydrate metabolism, protein folding/degradation, and DNA repair [83]. These results suggest that normal cells can antagonize aging by upregulating the transcription of Blm10, providing important insights into the mechanisms of aging and aging-related diseases.

### 3.2. PA200 Is Associated with Development of Certain Types of Tumors during Aging

Although cellular senescence inhibits tumor progression early in life, it is often accompanied by tumorigenesis [1,8,84]. The accumulation of large quantities of abnormal histone marks is a hallmark of both cellular senescence and tumorigenesis [85,86]. Later in life, some of the characteristics of senescent cells appear to mediate the development of age-related diseases, including cancer. Cellular senescence plays a crucial role in tumor genesis, progression, and metastasis [87]. Recently, PA200 has been shown to be highly expressed in non-small cell lung cancer (NSCLC), hepatocellular carcinoma, multiple myeloma, gastric cancer, esophageal squamous cell carcinoma, esophageal adenocarcinoma, oral squamous cell carcinoma (OSCC), and lung cancer [2,88,89,90,91,92,93,94,95]. This is consistent with the notion that PA200 inhibits cellular senescence. NSCLC is particularly well known for its ability to evade the immune system, proliferate, and metastasize throughout the body. NSCLC cells evade immune recognition by employing a variety of strategies, including reduced antigen presentation, increased expression of immunosuppressive molecules, and recruitment of immunosuppressive cells, eventually contributing to the resistance of NSCLC to immunotherapy [96,97]. Immune checkpoint inhibition (ICI) therapies have provided cures for many tumor patients. However, the response rate of patients with solid tumors to these therapies ranges from 10% to 50% [98]. Proteasomes play a critical role in various aspects of antitumor immunity, such as antigen processing and presentation, inflammation activation, and T-cell differentiation [99,100]. The immunoproteasome, activated by interferon, is also involved in these processes [101]. The expression of the immunoproteasome in tumors has been shown to correlate with the response to ICI therapy [102,103]. In contrast, high expression of PA200 decreases the response of many cancer patients to ICI therapy and patient survival [88]. Since PA200 is highly expressed in lung adenocarcinoma, where T cell-associated markers are downregulated, Aaron Javitt et al. proposed that PA200 may affect tumor progression by modulating T cell-mediated antitumor immunity [88]. MAPP (mass spectrometry analysis of proteolytic peptides) and proteasome profiling approaches revealed that the carboxy-terminal residue of the degraded peptides varied significantly between tumors and their adjacent samples. PA200 binds to both the constitutive 20S particle and the immunoproteasome 20S particle [22,23]. By comparing the classification of peptides that produce different amino acid residues after incubation with PA200 and each of the two types of 20S particles, PA200 inhibits the function of the immunoproteasome. This further explains the subsequent reduction in human leukocyte antigen (HLA) on the surface of A549 cells, which overexpress PA200. Accordingly, overexpression of PA200 reduced the diversity of peptides produced by the proteasome in lung adenocarcinoma cells [88]. PA200 overexpression reduces intracellular antigen processing and presentation in lung adenocarcinoma cells, inhibiting T-cell activity and conferring a ‘cold’ tumor phenotype [88]. Further animal studies demonstrated that PA200 regulates cytotoxic T cells in vivo, suggesting that PA200 might contribute to immune escape in NSCLC by reducing the activity of cytotoxic T cells (Table 1) [88].

In addition, PA200 not only modulates proteasome function but also has the potential to serve as a biomarker for a variety of malignancies. PA200 is predicted to be a therapeutic target because it is a proto-oncogene in gastric cancer [94]. PA200 is a biomarker for oral cancer [91]. Moreover, PA200 plays a crucial role in promoting hepatocyte regeneration [104]. Ge et al. discovered high PA200 expression in hepatocellular carcinoma (HCC) using the HCCDB and ONCOMINE databases [92]. Analysis of clinical data from the TCGA database revealed that patients with high PA200 expression had a significantly lower overall survival rate than those with low PA200 expression. The HCC tissue data from the TCGA-LIHC database were divided into two groups based on PA200 expression levels. Gene Set Enrichment Analysis (GSEA) was used to analyze the tumor-associated functional pathways of PA200. The analysis demonstrated that PA200 plays a significant role in the development of hepatocellular carcinoma by affecting phenotypic functional pathways such as cell proliferation, apoptosis, and the cell cycle. The related functional phenotypes were validated in PA200 knockdown cells [92]. Because PA200 delays senescence, as evidenced by the premature senescence phenotype observed in PA200 knockout mice [24], the increase of PA200 levels in tumor cells may promote tumor progression by delaying cellular senescence. GSEA analysis also revealed an association between the mTOR signaling pathway and the PA200-regulated HCC progression. Gene expression associated with the mTOR signaling pathway was analyzed. HepG2 cells with PA200 knockdown showed a significant reduction in the levels of phosphorylated mTOR compared to controls. Additionally, the protein levels of mTOR, Akt, MAPK, Erk, and Mek were also reduced. Furthermore, the RNA levels of both upstream and downstream components of the mTOR signaling pathway, including AMPK, Akt, c-myc, and PCNA, decreased after PA200 knockdown [92]. Perhaps PA200 directly degrades a suppressor of the mTOR pathway (Table 1) or indirectly downregulates its transcription following the transcription-coupled degradation of the core histones [24].

Glutamine serves as a precursor for nucleotide, protein, and lipid biosynthesis and promotes mTOR activity [105,106]. After ionizing radiation (IR) exposure, cells demonstrate an increased need for exogenous glutamine. Cells that contain PA200 can withstand this IR-induced glutamine demand, while cells that lack PA200 exhibit impaired long-term viability [106]. PA200-knockdown cells are unable to maintain intracellular glutamine levels. The radiosensitivity of PA200-knockdown cells can be reversed by additional glutamine supplementation. When extracellular glutamine is restricted, PA200-containing cells respond by slowing growth, but PA200-knockdown cells and cells in which the postglutamyl activity of proteasomes is inhibited are unresponsive and continue to grow rapidly. The levels of the mTOR substrate ribosomal S6 kinase (S6K) reflect the cellular unresponsiveness to nutrient depletion [107]. Therefore, the lack of available glutamine prevents the limitation of growth, leading to the continued growth and eventual death of PA200-deficient cells [107]. In conclusion, PA200 is important in maintaining glutamine homeostasis and is particularly crucial for the long-term survival of tumor cells after ionizing radiation exposure.

### 3.3. Degradation of Exogenous N-Terminal Fragment of Huntingtin Protein by PA200/Blm10-Proteasome

Aging is frequently related to various neurodegenerative diseases, such as Huntington’s disease (HD), which are often accompanied by abnormal protein aggregation. HD is a dominant, autosomal illness with gradual choreiform movements and progressive loss of speech, mobility, cognition, and swallowing abilities, probably caused by the accumulation of the aggregated abnormal huntingtin proteins with polyglutamines [108,109]. While PA200 binds the N-terminal fragment of huntingtin (N-Htt) with polyglutamines in vitro, the Blm10-proteasome can degrade the soluble N-Htt fragment in vitro [110]. Deletion of Blm10 or knockdown of PA200 increases the cellular levels of the exogenous N-Htt aggregates and cytotoxicity [110]. Further pathophysiological studies might clarify whether the PA200/Blm10-proteasome contributes to the maintenance of neuronal homeostasis by promoting the degradation of the abnormal huntingtin (Table 1).

Recently, the cold temperature (15 °C) has been shown to extend the life span of *Caenorhabditis elegans* by selectively inducing the trypsin-like activity of the proteasome by PSME3, the worm orthologue of human PA28γ/PSME3. Mechanistically, in the Huntington’s disease and amyotrophic lateral sclerosis (ALS) models of *C. elegans*, hypothermia-induced PA28γ reduced the aggregation of disease-associated proteins. Remarkably, a similar phenomenon was observed at 36 °C in cultured human cells [111], which not only suggests evolutionary conservatism in the regulation of proteasome activity at cold temperatures but also demonstrates the inextricable link between the proteasome and cellular senescence. Because PA200 binding to 20S CP also increases the trypsin-like activity of the proteasome, does the PA200 proteasome also have a role in slowing down senescence when induced at cold temperatures? This question may necessitate further research to answer.

### 3.4. PA200 Prevents Cellular Senescence in Mesenchymal Stem Cells

The Yes-associated protein (YAP) and the transcriptional coactivator with the PDZ-binding domain (TAZ) are critical regulators of tissue regeneration and stem cell circuitry by regulating stem cell renewal, fate, and plasticity [112]. YAP/TAZ is important in delaying stem cell senescence to prevent stem cell exhaustion [113]. PA200 depletes the nuclear acetylated YAP in the mesenchymal stem cells (MSC) treated with the histone deacetylase (HDAC) inhibitor apicidin. Injection of the PA200-knockdown MSCs into an infarcted heart supports that YAP depletion by PA200 in the nucleus is required for the maintenance of MSC therapeutic function in myocardial infarction [114]. Thus, PA200 might be important in promoting the differentiation of mesenchymal stem cells by reducing the levels of nuclear YAP. Further mechanistic studies might validate whether PA200 reduces nuclear YAP levels by promoting its proteasomal degradation or by regulating the cytosol–nucleus transport of the related proteins indirectly. Although PA200 prevents senescence in general by degrading the core histones with abnormal marks and maintaining the stability of the histone marks [24], it may also promote mesenchymal stem cell differentiation by downregulating anti-senescence proteins, such as YAP, in a special case (Table 1).

**Table 1 ijms-25-05637-t001:** PA200 plays an important role in aging-related diseases.

Aging-Related Disease		Mechanisms	References
Cancer	NSCLC	Reduces intracellular antigen processing and inhibits T-cell activity, leading to ICI therapy resistance.	[88]
	HCC	Activates mTOR signaling; increases Malignant Progression of HCC	[92]
Huntington’s disease		Decreases the cellular levels of the exogenous N-Htt aggregates	[110]
Myocardial infarction		Depletes YAP in the nucleus; promotes the cardiac commitment of MSC.	[114]

## 4. Conclusions and Perspectives for Future Studies

Loss of PA200 or its yeast ortholog Blm10 is the leading cause of the decline in proteasome activity during cellular senescence, whereas normal yeast cells might antagonize senescence by upregulating transcription of Blm10 [28]. The PA200-proteasome plays important roles in maintaining the stability of the histone marks, apparently by promoting histone degradation during transcription and senescence. PA200 deficiency accelerates aging in mice, leading to immune malfunction, anxiety, and a significantly shortened lifespan [24]. These senescence-associated phenotypes in PA200-deficient mice may be a result of the accumulation of “old” histones with aberrant histone marks. This anti-aging activity of PA200 is highly conserved evolutionarily since its yeast ortholog Blm10 functions similarly [28].

PA200 overexpression is associated with many tumors. PA200 overexpression in NSCLC might reduce the activity of the immunoproteasome and the variety of antigenic peptides [88], contributing to the resistance of NSCLC to ICI therapy. Upregulation of PA200 expression in HCC promotes hepatocellular carcinoma through the activation of the mTOR signaling pathway [92]. PA200 plays a crucial role in the survival of tumor cells after exposure to ionizing radiation by regulating cellular glutamine homeostasis [107].

In addition, PA200 binds the N-terminal fragment of huntingtin in vitro, and the Blm10-proteasome can degrade the soluble huntingtin fragment in vitro, though pathophysiological studies are required to clarify whether the PA200/Blm10 proteasome contributes to the maintenance of neuronal homeostasis by promoting the degradation of the abnormal huntingtin.

PA200 might be important in promoting the differentiation of mesenchymal stem cells by reducing the levels of YAP, but it is unclear whether PA200 reduces the YAP levels directly by promoting proteasomal degradation of YAP. Further studies on the validation of potential non-histone substrates of the PA200-proteasome, including huntingtin and YAP, might benefit the treatment of related neurodegenerative disorders, such as Huntington’s disease, and stem cell-mediated tissue regeneration, such as the treatment of myocardial infarction (Figure 2). Identification of other non-histone substrates of the PA200-proteasome would surely shed more light on the understanding of cellular senescence and related diseases.

## Figures and Tables

**Figure 1 ijms-25-05637-f001:**
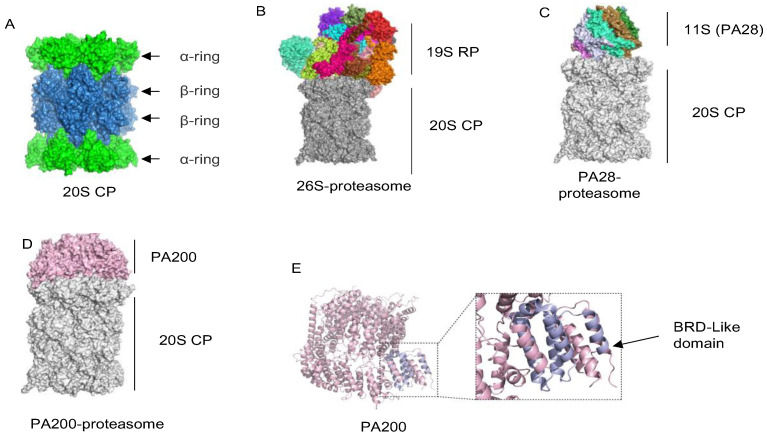
Structures of proteasomes. (**A**), 20S CP (PDB ID: 7PG9) (**B**), 26S proteasome (PDB ID:6MSH) (**C**), and the PA28 proteasome (PDB ID:7NAO) include the PA28αβ proteasome and the PA28γ proteasome. (**D**), PA200 proteasome (PDB ID: 6KWY) (**E**), the BRDL-domain of PA200 (PDB ID: 6KWX). Structural data from the PDB database.

**Figure 2 ijms-25-05637-f002:**
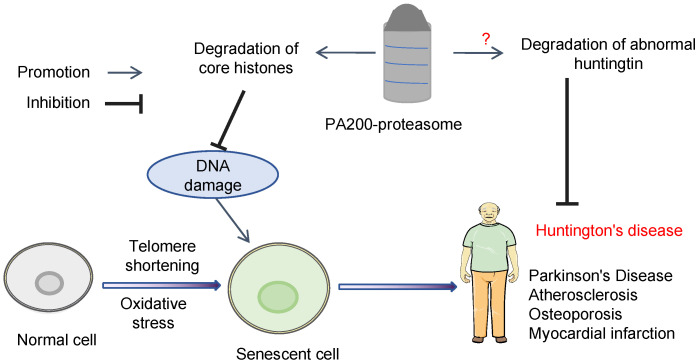
Scheme for the proposed mechanisms by which the PA200-proteasome regulates cellular senescence.

## Data Availability

Citations used in this article were identified using standard web data bases such as PubMed and PDB.

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
