# Peer review of "PA200-Mediated Proteasomal Protein Degradation and Regulation of Cellular Senescence"

_ijms, 2024, doi:10.3390/ijms25115637_

Round 1

Reviewer 1 Report

Comments and Suggestions for Authors

In the current review manuscript by Wen et al., the authors surveyed the role of mammalian proteosome activator, PA200-mediated proteasomal degradation of core histone proteins and associated their role in cellular senescence and various tumors. In the first half of the review article, they focused on the structural components of different proteosomes, their arrangement, and the roles of different subunits in substrate identification, and degradation. In the second part, they discussed the role of PA200 proteosome in preventing cellular senescence, their association with various tumors, and degradation of abnormal Huntington’s protein. Specifically, the authors highlighted the studies indicating that PA200 degrades the core histones with aberrant histone marks that regulate the expression of age-related genes, thus extending the lifespan. Similarly, they also discussed that the yeast homologue Blm10 reduces the H3 histone in the non-replicating G1-arrested cells. PA200 is also required for restricting the cellular growth upon the ionizing radiation treatment which induces the glutamate demand.

Major comments:

1.     The first section of the manuscript describing the structural models is not clear. It would greatly help the reviewers and subsequently the readers to correctly understand the structure of the various proteasome complexes discussed here. The authors should provide structural models for different proteosomes discussed here.

2.     Section 3.2 describing the association between PA200 with various tissues is not clear and confusing. There is lot of information here. The authors need to present this section in a more coherent manner.

3.     Section 3.2. The PA200 activity is reduced during senescence but elevated during various tumors. How do you correlate this ?

4.     How do you correlate the downregulation mTOR signaling pathway with PA200 depletion ?

5.     Does PA200 degrades abnormal Huntington proteins ? If not, the title of the section should be modified.

6.     YAP delays the senescence. If YAP is depleted by PA200 as described, higher levels of PA200 should be associated with early senescence ? Can you clarify this.

Minor comments:

1.     The authors do not expand multiple abbreviations when they introduce for the first time (CP, NSCLC, YAP)

2.     Line 72 – what are immunoproteasomes ?

3.     Line 149 – what are chaotic residues ?

4.     Line 344 – anti-aging

5.     Line 233 – full stop after the citations. Remove “a”

6.     There are spacing issues in the manuscript. Correct them.

Comments on the Quality of English Language

The authors can concise their descriptions at some parts of the manuscript. Further, they should edit and proofread the manuscript for grammatical errors. 

Author Response

Responses to Reviewer 1

Major comments:

  1. The first section of the manuscript describing the structural models is not clear. It would greatly help the reviewers and subsequently the readers to correctly understand the structure of the various proteasome complexes discussed here. The authors should provide structural models for different proteosomes discussed here.

RESPONSES: As the reviewer suggested, we have included the revised Figure 1 to illustrate the structures of the various types of proteasomes.

  1. Section 3.2 describing the association between PA200 with various tissues is not clear and confusing. There is lot of information here. The authors need to present this section in a more coherent manner.

RESPONSES: To improve the clarity of the section, we have included Table 1 to provide a concise summary of the relevant content.

  1. Section 3.2. The PA200 activity is reduced during senescence but elevated during various tumors. How do you correlate this?

RESPONSES: Thank you for bringing up this thought-provoking issue. Recent reviews on cellular senescence and tumor progression have shown a close relationship between them. It has been suggested that PA200 delays senescence, as evidenced by the premature senescence phenotype observed in PA200 knockout mice (ref.#24). Therefore, reduced levels of PA200 may facilitate senescence, but the increase of PA200 in tumor cells may promote tumor progression by delaying cellular senescence. This explanation has been included in the revised version (Page 7).

  1. How do you correlate the downregulation mTOR signaling pathway with PA200 depletion?

RESPONSE: Ge et al. showed that mTOR signaling is downregulated in the PA200-deficient HepG2 cells compared to wild-type cells (ref. #92). Perhaps, PA200 directly degrades a suppressor of the mTOR pathway or indirectly downregulates its transcription following the transcription-coupled degradation of the core histones (ref.#24 ). This explanation has been included in the revised version (Page 7).

  1. Does PA200 degrades abnormal Huntington proteins? If not, the title of the section should be modified.

RESPONSE: As suggested, we have changed the title from” Degradation of abnormal huntingtin protein by PA200/Blm10-proteasom” into “Degradation of exogenous N-terminal fragment of huntingtin protein by PA200/Blm10-proteasome” (Page 8)

  1. YAP delays the senescence. If YAP is depleted by PA200 as described, higher levels of PA200 should be associated with early senescence? Can you clarify this.

RESPONSE: Although PA200 depletes YAP in the nucleus, whether the specific mechanism is by promoting YAP degradation or by regulating transport is still unclear.  PA200 prevents senescence in general by degrading the core histones with abnormal marks and maintaining the stability of the histone marks (ref. #24), but it may also promote mesenchymal stem cell differentiation by downregulating anti-aging proteins, such as YAP, under a special case. This explanation has been included in the revised version (Page 8).

Minor comments:

  1. The authors do not expand multiple abbreviations when they introduce for the first time (CP, NSCLC, YAP)

RESPONSE: As suggested, we have now added the full names of CP (Page 5), NSCLC (Page 6), YAP (Page 8) and a list of abbreviations at the end of the text( Page 10).

  1. Line 72 – what are immunoproteasomes?

RESPONSE: We have now defined the immunoproteasome in the revised version (Page 2).

  1. Line 149 – what are chaotic residues?

RESPONSE: We have changed “chaotic residues” to “charged residues” (Page 4 ). 

  1. Line 344 – anti-aging

RESPONSE: Corrected as suggested..

  1. Line 233 – full stop after the citations. Remove “a”

RESPONSE: Corrected as suggested.

  1. There are spacing issues in the manuscript. Correct them.

RESPONSE: Corrected as suggested.

The authors can concise their descriptions at some parts of the manuscript. Further, they should edit and proofread the manuscript for grammatical errors.

RESPONSE: Thank you for your valuable feedback on the manuscript. We have carefully reviewed and made necessary revisions to the grammar of the article.

Reviewer 2 Report

Comments and Suggestions for Authors

The authors presented a manuscript that brings together current knowledge regarding the function of the proteasome activator PA200, which is profoundly linked to the regulation of cellular senescence. The choice of bibliographical references seems to me to be well made, allowing for an exhaustive and clear overview. Perhaps, I would have taken greater consideration of the most recent literature (2021-2024), specifically regarding this topic and I would have included some more explanatory schemes. The exposition in English seemed fluid and correct to me. My opinion is overall positive.

Author Response

RESPONSE: Thank you for your encouragement. Apparently, we did not explain well for certain recent literature. Thanks to your nice advice, we have now included more explanations to these papers. For examples, we have now expanded our discussions on the results from references related to mTOR (2022) and YAP (2023) (ref. # 92 # 114) in the revised version.

Reviewer 3 Report

Comments and Suggestions for Authors

The authors present a review of the current state of research on the PA200 proteasome activator and its involvement in the regulation of cellular senescence. 

The authors review an active and interesting field and the part of the review that deals with PA200 and histones/aging/Huntington’s is very well written, while the rest of the review has room for improvement (vide infra). There's no need for additional language editing.

I recommend publication after substantial revision.

Major issues:

Despite the focus on cellular senescence, the authors frequently mention other aspects of physiology such as spermatogenesis and devote substantial space to other proteasomal activators that are not the focus of this review. The field of proteasome regulation is huge and it is therefore important to provide an introduction that is accurate and at the same time highly focused. I find that the introduction and the part where the proteasome complexes are introduced could be streamlined and conceptualized, as this part at present contains many single sentence, single citation statements that that are not well connected and resemble in part notes rather than a finished text.

There are major issues with the illustrations, both on a technical, so to say, workmanship level and also on the fundamental level of content. Both figures are sloppily and carelessly assembled. As an example, in figure one, on the top panel, the label inside the shape that looks like a cartoon explosion has incorrect hyphenation leaving a stranded “e” on the second line. In the same part of the image on the right side, the label “telomere damage” has different font sizes which I assume is probably not intended. No effort went into the alignment of shapes or labels. The distance between arrows is inconsistent in both figures and in figure 2 for example arrows are superimposed on labelling making the labelling harder to read. The rightmost arrow on figure 2 looks absolutely ridiculous. It is also important to note that the image of the PA200-bound 20S proteasome shown in figure 2 is copied from the paper cited under #72. It may be necessary to request permission from the authors for reproduction. While I understand that graphic design is probably not the author's passion, I frankly find the level of carelessness that these figures have been assembled with to be condescending towards the reviewer and also to the potential reader. In the age of free software, were programs like GIMP or Inkscape are freely available, there is no excuse for figures that are this badly crafted. I am also very much convinced that a scientist should put sufficient effort into the preparation of the figures irrespective of the prestige of the journal they are intended for. While this journal may not be “Nature”, its readers certainly deserve better than this.

The figures are even more problematic on a different level. To this reviewer it is not clear at all what the figures are meant to say except maybe that, since this is a review and there is apparently space for figures, there have to be some figures. If we look at Figure 1 we see that apparently oxidative stress, DNA damage and telomere damage somehow via an arrow influence a vaguely flesh colored oval labeled “senescent cell”. This senescent cell is subdivided into a core which could be or could not be the nucleus - we don't know for sure - and several sections. It is unclear whether this is meant to be a pie chart or whether this is meant to show some kind of intracellular structure with weird radial subcellular compartments. If the latter is true, why would the nucleus be labeled “hallmark of senescent cell”? If it is some kind of pie chart then we wonder whether the size of the sectors matters. For example, if lysosome enzymes and beta-galactosidase are part of the same sector, does that mean that they are individually only half as important as let's say apoptosis resistance? In Figure 1 the arrows very obviously can mean very different things: the arrow on top means probably something like “causes” while the arrow on the right probably means something like “contributes to”. The arrows towards the tumor imply that the cell is really, actually physically changing into a tumor which is logically something entirely different. My impression is that Figure 1 could be best replaced by either a short table or maybe two or three well-crafted sentences. As a figure it fails on many levels and adds nothing to the review. Figure 2 is equally problematic: we see again the same entities oxidative stress, DNA damage, and telomere shortening, but this time affecting via an arrow the core histones, that then in turn are somehow affecting (or fed into?) the proteasome. Again, it is really not clear what the arrows are meaning and it's pretty obvious that the arrow between DNA damage and core histones must mean something different from the arrow between core histones and the proteasome. Strangely the proteasome somehow appears to be able to stop oxidative stress, DNA damage, or telomere shortening as an inhibitory arrow is drawn between the two. This again doesn't make much sense logically or biochemically, as we would not expect the proteasome to be somehow directly involved in control of chemical agents causing oxidative stress.

In their current state the figures add nothing to the review and only serve to confuse the reader. They are not adequate representations of the mechanisms that are correctly described in the review. As the authors use a substantial part of the paper to describe the architecture of the 20S proteasome core particle and the 19S regulatory particle as well as the PA28 regulatory particles, including also the immune proteasome and other specialized proteasomes, it may be more helpful to the reader to have a figure that shows an overview of these different protein complexes. Such a figure could show which of the subunits are common to all complexes and which of the subunits are defining specific subcomplexes. If this figure is crafted well, it could also replace a part of the full page that the authors are currently using to describe proteasome complexes that are not the focus of this review. As a wealth of structural biology data of the proteasome systems is available, these figures could be quite detailed and would add valuable information to the review. I understand that this review is mostly centered on biochemistry, cell biology and medical data, but laying a sound foundation of structure could improve it. It would also for example be interesting to see – more specifically - where the bromodomains of PA 200 that are mentioned in the review would be located in the context of the entire PA 200 proteasome complex.

Minor issues

Abbreviations are not always introduced. Some MDPI journals allow the addition of an index of abbreviations. If possible for this journal, this might be helpful.

Author Response

RESPONSES: According to your nice suggestions, we have made extensive corrections to our previous draft, the detailed corrections are listed below.

1, Comments: My impression is that Figure 1 could be best replaced by either a short table or maybe two or three well-crafted sentences.

RESPONSES: Thank you for your suggestion, we have changed the original Figure 1 to Table 1.

2, Comments: Figure 2 is equally problematic:

RESPONSES: According to your suggestion, we have changed the directions of arrows and the arrangement of the relevant elements. Thank you again for your patience in reviewing our manuscript.

3, Comments: it may be more helpful to the reader to have a figure that shows an overview of these different protein complexes

RESPONSES: Following your suggestion, we have added structures of various proteasomes in the revised Figure 1).

4, Comments: where the bromodomains of PA 200 that are mentioned in the review would be located in the context of the entire PA 200 proteasome complex.

RESPONSES: We have highlighted the location of BRDL domain in new Figure 1E.

5, Comments: Abbreviations are not always introduced.

RESPONSES: We have added a list of abbreviations at the end of the article.

Round 2

Reviewer 1 Report

Comments and Suggestions for Authors

Dear Authors, 

I appreciate for resolving all the comments in the revised version of the manuscript.

Author Response

Thank you very much for your comments, which have been of great help to us in improving the quality of the manuscript.

Reviewer 3 Report

Comments and Suggestions for Authors

I find that the table replacing Figure 1 is very useful compared to the previous figure and gives more information. I am happy to see that the authors have now included a figure that shows an overview of the proteasome complexes for which structures at present are available including a figure that points out the BRD-like domains in PA200. The new figure 1 is well crafted! It should be noted though that in panel C there should be a space between 11S and PA28; I would recommend fixing this before publication (in proof). All issues with figure 2 have been addressed and I find that the figure is now much more coherent. I commend the authors for having improved the figures in their paper to this extent.

I am grateful to the authors for addressing the issues pointed out by me. Their review is an excellent contribution for the field and deserves to be published.

Author Response

Thank you for pointing out our oversight, we have made the changes in Figure 1 (Page 4)